# AMOC Stability Amid Tipping Ice Sheets: The Crucial Role of Rate and Noise

Sacha Sinet[1,2], Peter Ashwin[3], Anna S. von der Heydt[1,2], and Henk A. Dijkstra[1,2]

[1]Department of Physics, Institute for Marine and Atmospheric research Utrecht, Utrecht University, Utrecht, The Netherlands
[2]Center for Complex Systems Studies, Utrecht University, Utrecht, The Netherlands
[3]Department of Mathematics and Statistics, University of Exeter, Exeter EX4 4QF, UK

**Correspondence:** Sacha Sinet (s.a.m.sinet@uu.nl)

**Abstract.** The Atlantic Meridional Overturning Circulation (AMOC) has recently been categorised as core tipping element, for it is believed to be prone to critical transition under climate change, implying drastic consequences on a planetary scale. Moreover, the AMOC is strongly coupled to polar ice sheets via meltwater fluxes. On one hand, most studies agree on the fact that a collapse of the Greenland ice sheet would result in a weakening of AMOC. On the other hand, the consequences of a collapse of the West Antarctica ice sheet are less well understood. However, some studies suggest that meltwater originating from the Southern Hemisphere is able to stabilize the AMOC. Using a conceptual model of the AMOC and a minimal parameterization of ice sheet collapse, we investigate the origin and relevance of this stabilization effect in both the deterministic and stochastic cases. While a substantial stabilization is found in both cases, we find that rate and noise-induced effects have substantial impact on the AMOC stability, as those imply that leaving the AMOC bistable regime is neither necessary nor sufficient for the AMOC to tip. Also, we find that rate-induced effects tend to allow for a stabilization of the AMOC in cases where the peak of the West Antarctica ice sheet meltwater flux occurs before the peak of the Greenland ice sheet meltwater flux.

## 1 Introduction

Several components of the Earth system might abruptly and/or irreversibly collapse under climate warming. These are known as tipping elements (Lenton et al., 2008), each associated to a tipping point - a critical value of forcing above which the system is going to collapse. In a recent assessment, Armstrong McKay et al. (2022) classified the Atlantic Meridional Overturning Circulation (AMOC), the Greenland Ice Sheet (GIS) and West Antarctica Ice Sheet (WAIS) as global core tipping elements, as the loss of any of those climate components would lead to severe consequences on a planetary scale. There, it was also suggested that those three tipping elements might be at risk within the Paris agreement warming range, while both polar ice sheets might already have crossed their critical threshold of global warming. Furthermore, an ongoing destabilization is in general supported by observations (see Caesar et al. (2018) for the AMOC, Shepherd et al. (2018) for the WAIS and Shepherd et al. (2020) for the GIS) or studies based on early warning signals (see Boers (2021), Michel et al. (2022), Ditlevsen and

Ditlevsen (2023) and van Westen et al. (2024) for the AMOC, Rosier et al. (2021) for the WAIS and Boers and Rypdal (2021) for the GIS).

From a theoretical standpoint, tipping events are most often understood as the expression of a dynamic bifurcation intrinsic to the system. In this case, bifurcation-induced tipping can be avoided ensuring that external drivers (typically the global temperature) do not cross the critical value at which the bifurcation takes place. However, tipping events can also occur via inherently different processes, as described by Ashwin et al. (2012). On one hand, those tipping events can be induced without crossing any critical value of external forcing and, in fact, without the presence of any bifurcation. Such tipping phenomena typically occur when the rate of variation of external drivers is faster than the rate at which the system relaxes back to its attractor. The potential impact of those rate-induced tipping phenomena on the AMOC stability has been pointed out by numerous studies involving conceptual (Alkhayuon et al., 2019; Ritchie et al., 2023; Klose et al., 2023) or global ocean models (Lohmann and Ditlevsen, 2021). On the other hand, a tipping event can be triggered by the presence of stochastic noise, routinely used to represent the unresolved variability of processes acting on a shorter timescale (Hasselmann, 1976; Frankignoul and Hasselmann, 1977). In this case, a system can collapse before crossing any critical value of forcing, solely due to the noise. In fact, noise-induced tipping phenomena potentially explain abrupt transitions observed in paleoclimatic records known as Dansgaard-Oeschger events (Ditlevsen et al., 2007; Ditlevsen and Johnsen, 2010). The impact of noise on the AMOC stability has been investigated in a conceptual model by Castellana et al. (2019). There, the variability associated to the gradient of freshwater forcing over the Atlantic Ocean is able to induce a collapse of the overturning within the next century, with a probability estimated at about 15%. In a later study, similar noise-induced tipping events have been shown to also occur in an ensemble of CMIP5 models (Castellana and Dijkstra, 2020).

As a further complication, the study of each tipping element individually might not suffice for producing reliable projections of tipping behaviour. Indeed, those elements are not isolated but may interact with each other, sometimes strongly, making it harder to predict the occurrence and severity of a tipping event. Arguably, the AMOC, GIS and WAIS form the most prominent example of such a coupled system of tipping elements. All three interact through various processes and on different timescales (Wunderling et al., 2021, 2024). On one hand, a GIS collapse and the associated release of meltwater into the northern Atlantic Ocean is known to imply, at best, a weakening of the AMOC (Jackson and Wood, 2018; Bakker et al., 2016). On the other hand, a collapse of the AMOC could imply a warming of the Southern Hemisphere (Jackson et al., 2015; Stouffer et al., 2006), thereby accelerating ice loss in the WAIS (Favier et al., 2019; Joughin et al., 2014). Such chain of destabilizing events are often referred to as cascading tipping, or domino effect (Dekker et al., 2018; Klose et al., 2021; Wunderling et al., 2021, 2024). However, this coupled system might also contain some stabilizing feedback. In this case, the collapse of one component may stabilize others, thereby making a global cascading event less likely. For example, using the state-of-the-art climate model HadGEM3, the collapse of the AMOC has been shown to imply a cooling of typically 5 to 8°C over Greenland (Jackson et al., 2015), which would inhibit ice loss of the GIS.

A substantial source of uncertainty in how cascading tipping events could unfold lies in how a sudden ice loss from the WAIS would impact the AMOC. There, many different processes are at play and might either tend to destabilize it or keep it from collapsing (Swingedouw et al., 2009; Berk et al., 2021). In a conceptual model representing the AMOC and both polar

ice sheets, Sinet et al. (2023) recently showed that the WAIS meltwater flux is able to prevent an AMOC collapse against the destabilization caused by both climate warming and ice loss from the GIS. In this study, the occurrence of an AMOC collapse drastically depends on the rate of forcing, as well as on the time delay between the tipping of both ice sheets. However, rate-induced effects were not explored in detail, and the stochastic case was not considered.

Motivated by the WAIS induced stabilization effect on the AMOC found by Sinet et al. (2023), we explore its relevance in a more comprehensive conceptual model of the AMOC, considering also the stochastic case. For this, we perform a variety of model simulations using a simplified representation of tipping ice sheets. We aim to identify the distinct roles of rate, noise and bifurcation-induced effects. The experimental setup is presented in section 2. In section 3, we reproduce a set of model simulations similar to Sinet et al. (2023), yielding a qualitatively similar AMOC stabilization effect of the WAIS meltwater flux. In section 4, the bifurcation structure of the model is presented, allowing us to identify the presence of rate-induced effects. The robustness of this stabilization effect in the stochastic case is investigated in section 5. Finally, a summary and a discussion are provided in section 6.

## 2 AMOC Model and Parameterized Ice Sheet Collapse

In this section, we describe the conceptual model used to represent the AMOC, and introduce a minimal representation of meltwater fluxes originating from the collapse of the GIS and WAIS.

The AMOC is represented using the model introduced by Cimatoribus et al. (2014). There, the Atlantic Ocean is divided into five different regions depicted by boxes having distinct salinity and whose volume are allowed to vary depending on the pycnocline depth (see Fig. 1). This box model includes two thermocline boxes t and ts, where the latter represents a region in which the isopycnal slopes are greater than in other parts of the thermocline and allows for computing the density gradient within the Atlantic basin. Furthermore, the model is forced via two surface freshwater fluxes $E_s$ and $E_a$ (represented through virtual salt fluxes). The former stands for the amount of evaporation occurring in the tropical region and equally redistributed to each pole, while the latter introduces a pole-to-pole gradient of freshwater forcing. Given a value of $E_s$, Cimatoribus et al. (2014) have shown that this model can yield bistability within some interval of $E_a$. Within this interval, one branch of stable steady states corresponds to a present-day-like AMOC with downwelling of dense waters in the northern box, or positive overturning strength ($q_{\mathrm{n}} > 0$), while the other corresponds to a ceased overturning, or vanishing overturning strength ($q_{\mathrm{n}} = 0$). Hereafter, we call those respectively ON and OFF state. For a more detailed description of the AMOC model and numerical methods, see appendix A.

To mimic the meltwater flux originating from collapsing ice sheets, time-dependent meltwater fluxes $F_{\mathrm{N,S}}(t)$ (also represented through virtual salt fluxes) are inserted in the northern and southern box respectively, and compensated for via evaporation at the equator (see Fig. 1). Those are assumed to yield a parabolic shape in time, as follows,

$$F_{\mathrm{N}}(t; P_{\mathrm{N}}) = \begin{cases} -\dfrac{6V_{\mathrm{N}}}{P_{\mathrm{N}}^3} t(t - P_{\mathrm{N}}) & \text{if } 0 < t < P_{\mathrm{N}}, \\ 0 & \text{otherwise,} \end{cases} \tag{1}$$

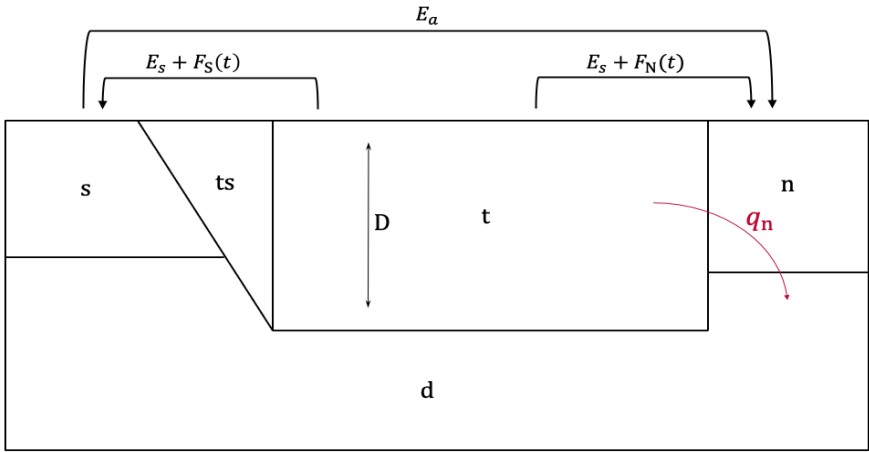

**Figure 1.** The AMOC dynamics is captured by water transport between five distinct boxes, namely the Southern Ocean box s, two thermocline boxes t and ts, a deep ocean box d and the northern Atlantic box n. Each individual salinity is allowed to vary, as well as the pycnocline depth D. The parameters $E_a$ and $E_s$, denoting the asymmetric and symmetric component of the surface freshwater flux respectively, are chosen to initialise the AMOC in an ON state (i.e. with strictly positive overturning strength $q_n$). Meltwater fluxes $F_{N,S}(t)$ are added to the northern and southern box respectively, and compensated for via evaporation at the equator.


$$F_S(t; P_S, \Delta t) = \begin{cases} -\dfrac{1}{4}\dfrac{6V_S}{P_S^3}(t - \Delta t)(t - \Delta t - P_S) & \text{if } 0 < t - \Delta t < P_S, \\ 0 & \text{otherwise,} \end{cases} \tag{2}$$

where $t$ is the time (in years), and $P_{N,S}$ is the duration of the GIS and WAIS collapse (also in years), respectively. Given that the GIS collapse is always initiated at $t = 0$, the parameter $\Delta t$ is the time at which the WAIS collapse is initiated, or the time delay between the initiation of those two tipping events (which can be both positive or negative). Also, the parameters $V_N = 2.99 \times 10^{15}$ m$^3$ and $V_S = 3.39 \times 10^{15}$ m$^3$ are the volumes of respectively the GIS and WAIS (as in Sinet et al. (2023), based on Morlighem et al. (2017, 2020)). To account for the WAIS meltwater lost in the Pacific or Indian Oceans, we consider only a quarter of the meltwater originating from the WAIS to reach the Atlantic Ocean, as in Sinet et al. (2023). An alternative representation of $F_S(t)$ will also be used, where the tipping delay $\Delta t$ is replaced by the delay between the maxima of both metlwater fluxes given by $\Delta t_{\max} = \Delta t + \frac{P_S - P_N}{2}$, which is positive if the maximum of $F_S(t)$ occurs after the maximum of


$F_{\mathrm{N}}(t)$. In this case, equation (2) is replaced by

$$F_{\mathrm{S}}(t; P_{\mathrm{S}}, \Delta t_{\max}) = \begin{cases} -\dfrac{1}{4}\dfrac{6V_{\mathrm{S}}}{P_{\mathrm{S}}^3}\left(t - \Delta t_{\max} + \dfrac{P_{\mathrm{S}} - P_{\mathrm{N}}}{2}\right)\left(t - \Delta t_{\max} - \dfrac{P_{\mathrm{S}} + P_{\mathrm{N}}}{2}\right) & \text{if } 0 < t - \Delta t_{\max} + \dfrac{P_{\mathrm{S}} - P_{\mathrm{N}}}{2} < P_{\mathrm{S}}, \\ 0 & \text{otherwise.} \end{cases}$$

(3)

Hence, the forcing (1)-(2) (or equivalently (1)-(3)) allows for conceptually capturing a full collapse of both ice sheets while spanning a wide range of possible tipping durations and delays, which is motivated by the uncertainty in ice sheet tipping points, tipping timescales and global warming trajectories (Armstrong McKay et al., 2022), as well as the important role of
those parameters found in Sinet et al. (2023). In the remainder of this document, we consider the interval $[50, 5000]$ years for the WAIS tipping duration $P_{\mathrm{S}}$, encompassing about the lower half of the interval $[500, 13000]$ years proposed by Armstrong McKay et al. (2022). Also, to explore the impact of different ice sheet collapse trajectories, we use the delay interval $[-1500, 1500]$ years for both $\Delta t$ and $\Delta t_{\max}$.

## 3   Deterministic Regime

In this section, we perform a set of model simulations to highlight the existence of a stabilizing impact of WAIS meltwater fluxes on the AMOC. For this, we initialise the AMOC in an ON state, choosing $E_a = 0.31$ Sv. First, we investigate the situation in which only meltwater originating from the GIS is introduced in the Atlantic Ocean. In this case, the forcing is given by equation (1) alone and is fully determined by the parameter $P_{\mathrm{N}}$. We find that for $P_{\mathrm{N}} \leq 1284$ years, the AMOC ends in an OFF state. In the critical case $P_{\mathrm{N}} = 1284$ years, the OFF state is reached at about year $850$ as the meltwater flux from the GIS
amounts to approximately $0.10$ Sv, consistent with typical forcing values leading the AMOC to a weak state in comprehensive models (Jackson and Wood, 2018).

To check whether stabilization effects associated to WAIS meltwater fluxes exist in this model, we force the AMOC with meltwater from both ice sheets. For this, we fix $P_{\mathrm{N}} = 1000$ years in equation (1) such that, by virtue of the previous experiment, the GIS forcing alone is sure to drive the AMOC to collapse. Indeed, using this value of $P_{\mathrm{N}}$, the OFF state is reached at
about year $330$ and the meltwater flux from the GIS reaches a maximal value of $0.14$ Sv (see Fig. 2.a). The WAIS melting trajectory is fully determined by either $(\Delta t, P_{\mathrm{S}})$ via equation (2) or $(\Delta t_{\max}, P_{\mathrm{S}})$ via equation (3). The result of experiments involving meltwater fluxes from both ice sheets is represented in Fig. 2.c in the $(\Delta t, P_{\mathrm{S}})$ parameter space and in Fig. 2.d in the $(\Delta t_{\max}, P_{\mathrm{S}})$ parameter space. We find a closed region in which the AMOC remains in an ON state (inside the dark blue boundary), hereafter called the stabilization region, while an OFF state was reached everywhere else (outside of the dark blue
boundary). In particular, we find that the stabilization region exists for realistic values of the WAIS tipping duration $P_{\mathrm{S}}$ (i.e. above the grey dashed line in Fig. 2.c-d).

The existence of a stabilization region can be understood as explained in Cimatoribus et al. (2014). Namely, given the sense of the circulation, meltwater inserted into the southern box implies a freshening of the box ts (see Fig. 1). This results in an increase of the density gradient between box ts and box n, and hence an increase of the overturning strength $q_{\mathrm{n}}$ (see appendix

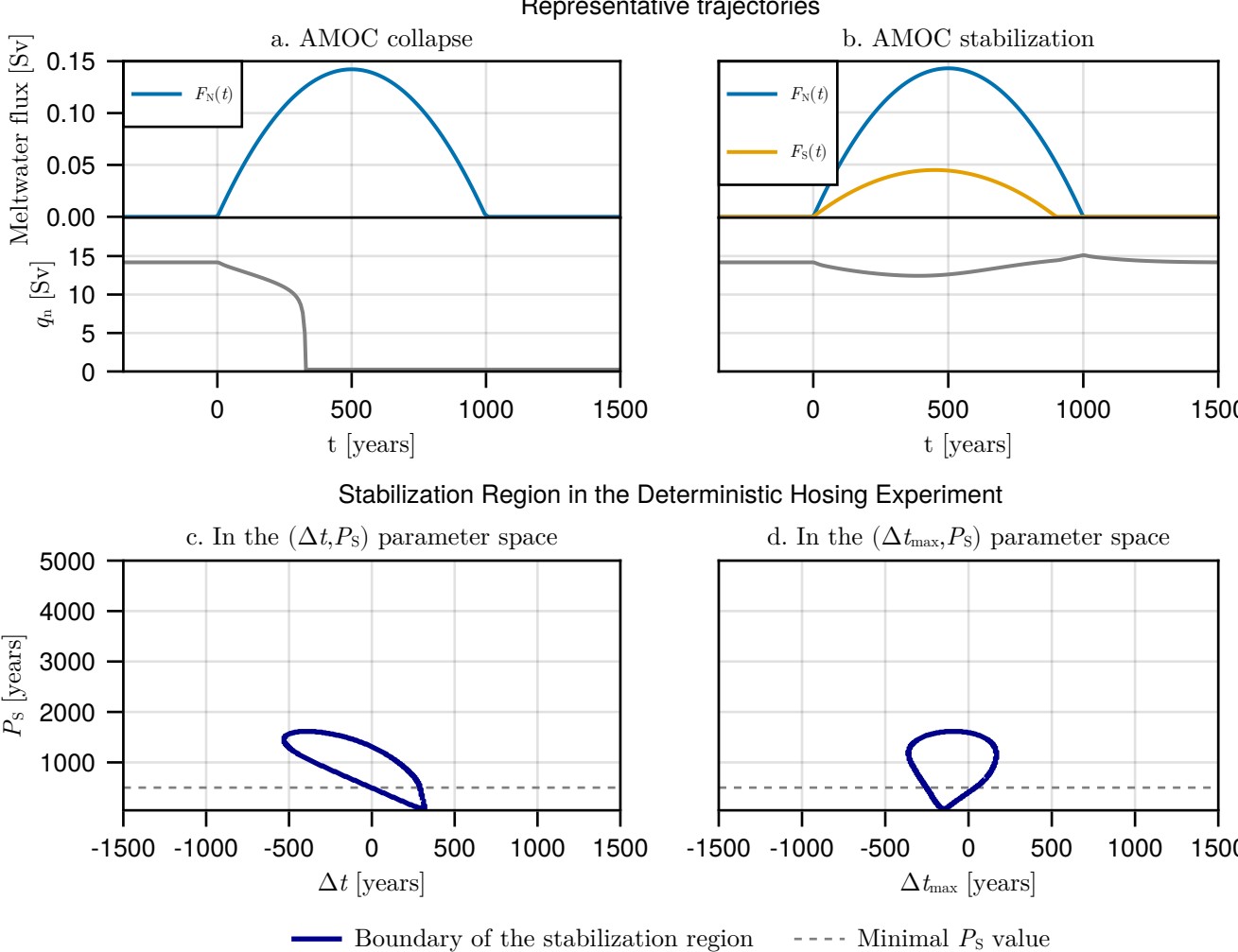

**Figure 2.** (a-b) Representative trajectories of meltwater fluxes $F_{N,S}(t)$ as given by the set of equations (1)-(2) (or equations (1)-(3)) and overturning strength $q_n$. In (a), only a GIS collapse lasting $P_N = 1000$ years forces the AMOC model, resulting in an AMOC tipping ($q_n = 0$). In (b), both a GIS and a WAIS collapse force the AMOC model. Those last $P_N = 1000$ years and $P_S = 900$ years, respectively, and are initiated at the same time ($\Delta t = 0$ years, or equivalently $\Delta t_{max} = -50$ years), resulting in an AMOC stabilization. (c-d) Boundary of the stabilization region in deterministic experiments involving both ice sheets, where the duration of the GIS collapse is fixed to $P_N = 1000$ years. Those are shown in (c) the $(\Delta t, P_S)$ parameter space and (d) the $(\Delta t_{max}, P_S)$ parameter space. All parameter configurations within the stabilization region (dark blue curve) preserve the AMOC in an ON state ($q_n > 0$), while those outside lead the AMOC to an OFF state ($q_n = 0$). The grey dashed line represents the minimal value of the WAIS tipping duration at $P_S = 500$ years proposed by Armstrong McKay et al. (2022).

A). However, in Fig. 2.c, there appears to be an upper limit of tipping delay at $\Delta t \approx 320$ years above which no stabilization can take place. Indeed, if no meltwater flux from the WAIS has already been applied by this time, the AMOC is already engaged in its collapse. Also, the overall shape of the stabilization region indicates that, to result in an efficient stabilization of the AMOC, a longer WAIS collapse has to happen earlier than a shorter one. Although in a different experimental setup, these results qualitatively agree with those in Sinet et al. (2023). In both Fig. 2.c and 2.d, there is an upper limit of WAIS tipping duration at $P_S \approx 1600$ years, above which $F_S(t)$ does not reach sufficiently high values to compensate for the comparatively strong GIS meltwater flux. Finally, in Fig. 2.d, the approximate symmetry of the stabilization region suggests the existence of a negative optimal value of $\Delta t_{\max} \approx -150$ years. This value is optimal in the sense that the range of $P_S$ values for which a stabilization occurs shrinks as $\Delta t_{\max}$ moves away from it, until the border of the stabilization region is reached. In other words, stabilization is most likely if the maximum of the WAIS meltwater flux occurs about 150 years before the maximum of the GIS meltwater flux.

Those results can be interpreted in the context of future climate change. In Armstrong McKay et al. (2022), plausible values for the GIS tipping duration $P_N$ are given in the range $[1000, 15000]$ years, with a most likely value of 10000 years. In our model, this most likely scenario results in the AMOC to remain stable regardless of the applied WAIS meltwater flux. Instead, $P_N = 1000$ years can be thought of as a worst case scenario. This is further motivated by the modelling study of Aschwanden et al. (2019), in which a GIS collapsing on the millennial timescale was found under a RCP8.5 scenario. Under such conditions, the whole range of plausible WAIS tipping points of $[1.5, 3.0]°C$ of global warming above pre-industrial levels (Armstrong McKay et al., 2022) would be overshot in less than a century, resulting in a negligible value of the time delay between ice sheet tipping events ($\Delta t \approx 0$ years). Hence, in such a worst case scenario, a stabilization would occur for values of $P_S$ between approximately 500 and 1300 years (see the representative trajectory in Fig. 2.b).

## 4 Bifurcation Structure and Rate Induced Effects

To gain a better understanding of the stabilization effect found in Fig. 2, we inspect the bifurcation structure of the AMOC model computed via numerical continuation using the Julia package BifurcationKit.jl (Veltz, 2020). In Fig. 3.a, steady states of the model are shown in terms of the overturning strength $q_n$ versus positive values of the meltwater flux in the northern box $F_N$. Those have been computed for three different values of $F_S = 0.00, 0.04$ and $0.08$ Sv, yielding three distinct curves (with $F_S$ increasing from darker to lighter). In each case, bistability is observed within some interval of $F_N$ above which the branch of stable present-day-like AMOC ceases to exist, while the collapsed AMOC configuration ($q_n = 0$) is always stable. This is sufficient to infer the existence of a bifurcation-induced tipping point, implying the potential of a critical transition as $F_N$ exceeds it. Via a similar approach, we also find that the sensitivity of the overturning strength $q_n$ to WAIS meltwater fluxes in the reference ON state is higher but comparable to what was found in Li et al. (2023) using the coupled climate model GISS-E2.1-G (see section S1). We note the presence of Hopf bifurcations which move along the unstable branch as $F_S$ increases, and a Bogdanov-Takens bifurcation creating a new Hopf bifurcation on the stable branch, which keeps close to the saddle-node bifurcation observed for each value of $F_S$ (see Fig. 3.b). However, in terms of how the southern meltwater flux impacts the

stability landscape, the most important general feature observed here is the expansion of the bistable region to higher values of $F_N$ when $F_S$ increases. This can in principle explain the stabilization effect taking place in previous experiments, as the tipping point of the AMOC becomes harder to reach.

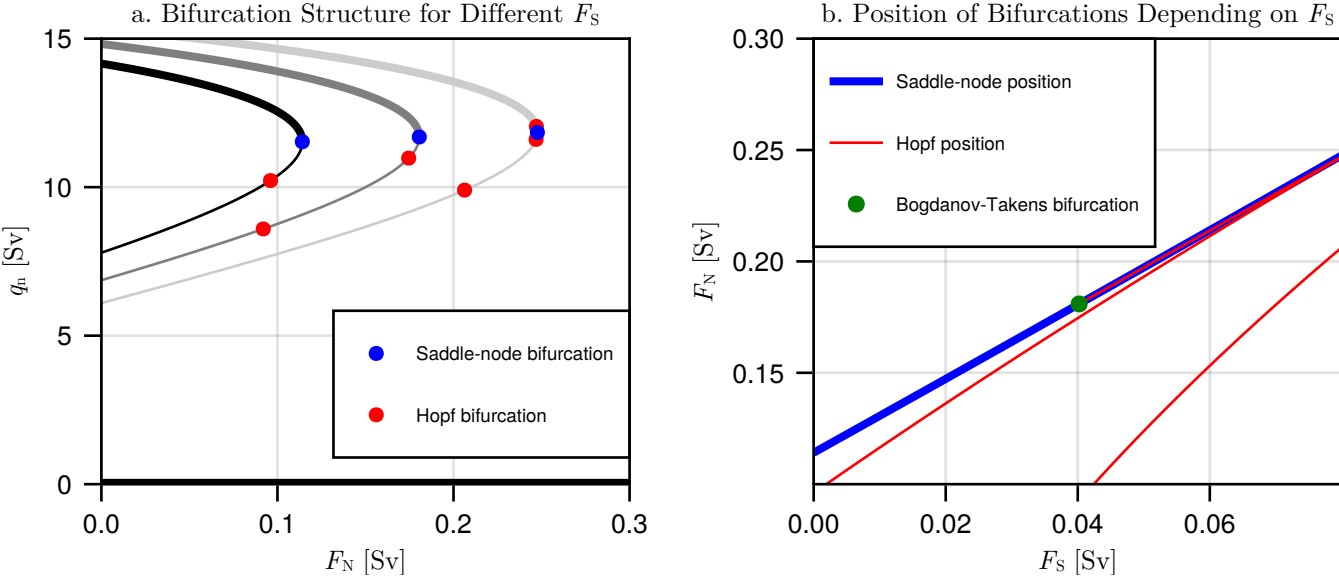

**Figure 3.** (a) Bifurcation structure of the AMOC model in the $(q_n, F_N)$ space. Thicker branches stand for stable steady states, while thinner branches indicate unstable steady states and bifurcations are denoted by coloured points. The bifurcation structure has been computed for three different values of $F_S = 0.00$, $0.04$ and $0.08$ (darker to lighter) . (b) Displacement of bifurcations as $F_S$ varies. To the right of the Bogdanov-Takens bifurcation (green), a new Hopf bifurcation is created (in red, superposed to the saddle-node bifurcation).


The value of $F_N$ at which the bistable region ends is approximately proportional to $F_S$, with a proportionality coefficient $\delta$ approximately given by $1.68$ (Fig. 3.b). Noting that in the case where $F_S = 0.00$ Sv, the bistable region ends at $F_{N,c} \approx 0.11$ Sv, it is possible to identify meltwater flux trajectories that never lead the AMOC outside of its bistable region. Namely, those satisfy

$$F_N(t) - \delta F_S(t) < F_{N,c}, \tag{4}$$

for all $t \in \mathbb{R}$. Given a fixed value of $P_N$, these form a closed and compact region in the $(\Delta t, P_S)$ and $(\Delta t_{max}, P_S)$ parameter space, for which the mathematical expression is given in appendix B. Hence, fixing $P_N = 1000$ years as in the experiment represented in Fig. 2.c-d, we can identify all parameter values for which no tipping point is ever overshot, hereafter called the no-overshoot region. In Fig. 4.a-b, the limits of the no-overshoot region are drawn in red, and the boundary of the stabilization
region found in Fig. 2 is drawn in dark blue. We find that these regions do not coincide, meaning that the AMOC stability cannot be predicted based on its bifurcation structure. Especially, we find that a collapse of the AMOC can happen inside the no-overshoot region (for parameter configurations within the red contour but outside of the dark blue contour). In this case, the

AMOC reaches an OFF state without ever leaving its bistable region. Also, we find that an AMOC tipping avoidance (i.e. a stabilization) can occur even in some cases where the tipping point is overshot (for parameter configurations within the dark
blue contour but outside of the red contour), a phenomena known as safe overshoot (Ritchie et al., 2021). Those are rate-induced effects, resulting in rate-induced tipping (Ashwin et al., 2012) or tipping avoidance (Ritchie et al., 2023), respectively.

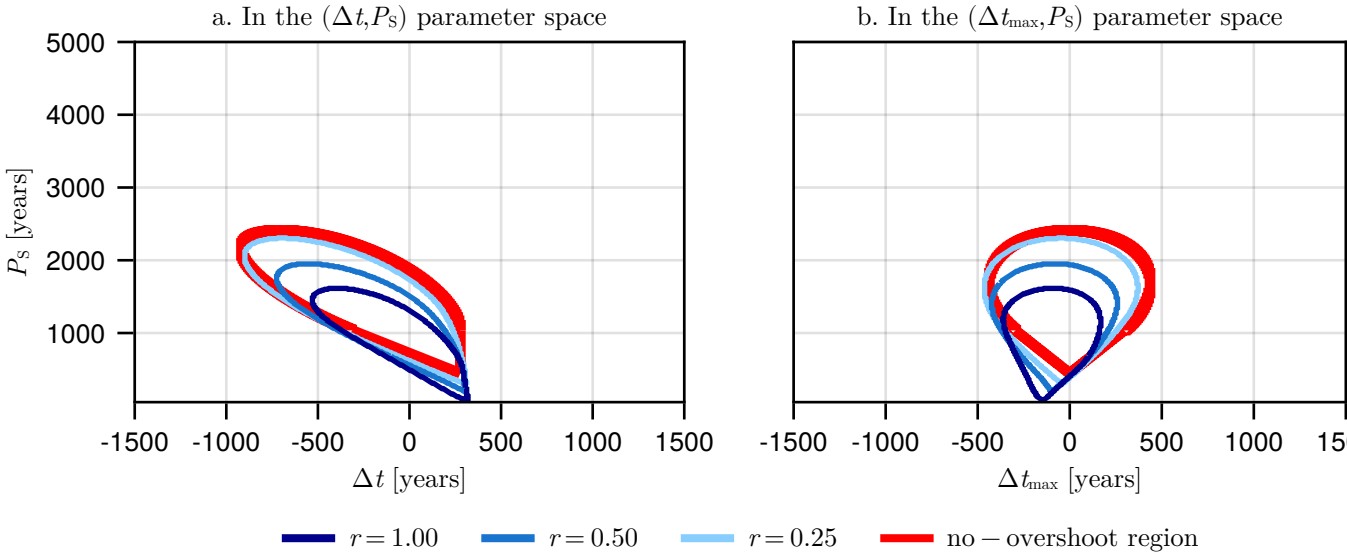

Figure 4. Boundary of the no-overshoot region (in red) and the stabilization region computed for different values of the rate parameter $r$ (nuances of blue), drawn in (a) the $(\Delta t, P_\mathrm{S})$ parameter space and (b) the $(\Delta t_\mathrm{max}, P_\mathrm{S})$ parameter space. The darker blue contour ($r = 1$) is the same as the one presented in Fig. 2.c-d, while lighter blue contours correspond to lower values of $r = 0.50$ and $0.25$.

Rate-induced effects are less pronounced as the rate at which the forcing varies is decreased. This can be tested via artificially expanding the forcing timescale, using $F_{\mathrm{N,S}}(rt)$ rather than $F_{\mathrm{N,S}}(t)$ in the experiments. This has the effect of multiplying durations and delays by the inverse of the rate parameter $r$. In other words, the rate of variation of the forcing decreases as
$r$ decreases. For values of $r$ lower than one, we stray away from reality as bigger volumes of meltwater are inserted in the Atlantic Ocean, but the bifurcation structure of the AMOC model under constant forcing remains unchanged, as well as the shape of the no-overshoot region. In Fig. 4.a-b, we represent the boundary of the stabilization region in the same experiment using $r = 0.50$ and $0.25$ (lighter tones of blue). As expected, we find that the no-overshoot region (delimited by the red curve) gradually becomes a better approximation of the stabilization region, translating into a suppression of rate-induced effects. This
results in transitions that are more consistent with the bifurcation-induced tipping case. Finally, we explain the approximate symmetry of the stabilization region in the $(\Delta t_\mathrm{max}, P_\mathrm{S})$ parameter space found in Fig. 2.d. The no-overshoot region (bounded

by the red curve in Fig. 4.b) is perfectly symmetric around $\Delta t_{\max} = 0$, which can also be directly inferred from its mathematical expression (see appendix B). Hence, in the limit where $r$ reaches zero, $\Delta t_{\max} = 0$ is the optimal value to result in a stabilization. While it is clear that the symmetry is broken as the value of $r$ increases to one, the stabilization region still preserves some
degree of symmetry around some optimal, negative value of $\Delta t_{\max}$. In other words, the (negative) optimal value of $\Delta t_{\max}$ depends on the rate of forcing.

The importance of rate-induced effects in this model can be explained by the fact that trajectories are relatively slowly attracted to the stable attractor. As an illustration, we note that the previously found critical value of GIS tipping duration at $P_{\mathrm{N}} = 1284$ years in section 3 does not result in an overshoot of the tipping point. In this case, performing the same simulation
with decreased $r$ results in the AMOC to never reach the OFF state (as shown in Fig. 5.a). This originates from the slow timescale induced by the dynamics of the pycnocline depth D. In fact, the decay time towards the stable equilibrium can be computed as the inverse of the least negative eigenvalue of the associated stable state which, as shown in Fig. 5.b, is always above 190 years. This slow timescale can be connected to the pycnocline depth by computing the associated eigenvector, in which we find that the direction corresponding to the pycnocline depth D dominates the others by approximately a factor 100.
This indicates that, for the values of $P_{\mathrm{N}}$ considered, the separation between the forcing timescale and the decay timescale toward stable equilibria (driven by the dynamics of the pycnocline depth) is not large enough to prevent rate-induced effects.

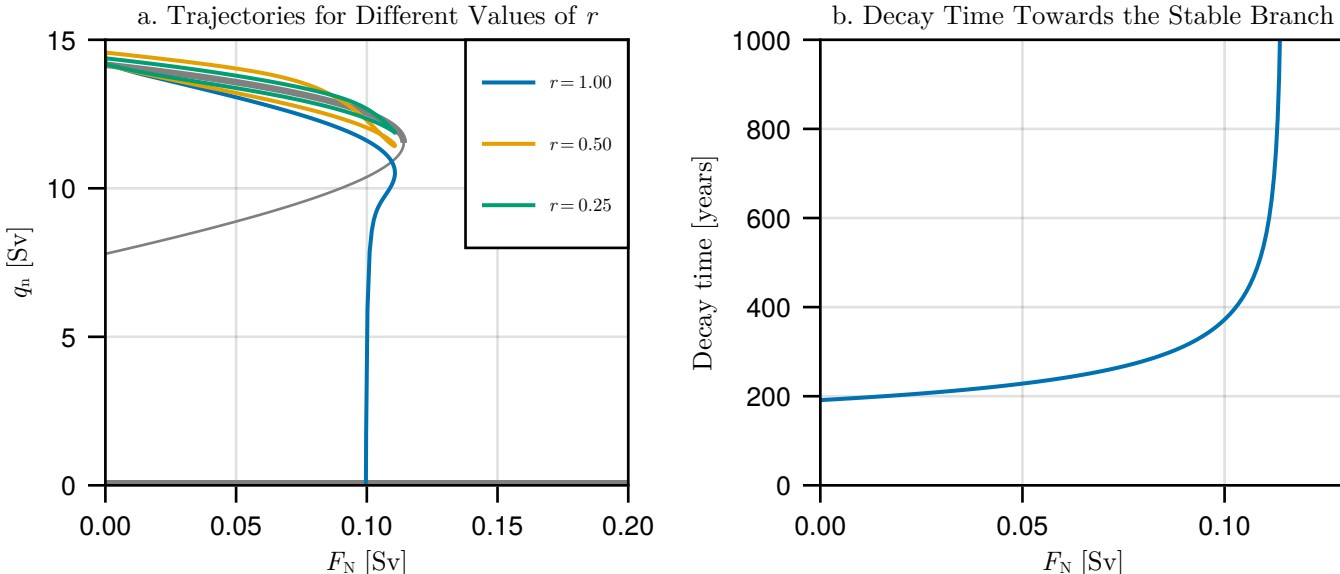

**Figure 5.** (a) Trajectories in the experiment where only the GIS meltwater flux forces the AMOC via equation 1, using $P_{\mathrm{N}} = 1284$ years and represented in the $(q_{\mathrm{n}}, F_{\mathrm{N}})$ space for different values of the rate parameter $r = 1.00, 0.50$ and $0.25$. Steady states for fixed $F_{\mathrm{N}}$ are added in grey, where thicker lines stand for the stable states. (b) Variation of the decay time along the upper stable branch. The minimum is for $F_{\mathrm{N}} = 0.00$ at about 190 years.

## 5 Weak Noise Regime

In this section, we evaluate the relevance of the stabilization effect of WAIS meltwater fluxes on the AMOC when natural variability is added to the surface freshwater flux. To do so, we redefine the asymmetric component of the surface freshwater flux $E_a$ in the following way (Castellana et al., 2019; Jacques-Dumas et al., 2023)

$$E_a(t) = \bar{E}_a(1 + f_\sigma \xi(t)). \tag{5}$$

Here, $\bar{E}_a$ is a constant representing the mean value of the asymmetric component of the surface freshwater flux, $\xi(t)$ is a white noise process with zero mean and unit variance, and $f_\sigma$ is the noise strength parameter. As found by Castellana et al. (2019), a non-zero value of $f_\sigma$ renders noise-induced tipping possible, where the AMOC can reach an OFF state solely due to the noise, independently of any meltwater forcing. From this point of view, both the cases in which the tipping point of the AMOC model can be overshot or not are relevant, and will be considered. To choose the noise strength parameter $f_\sigma$ we note that, at the value $\bar{E}_a = 0.31$ (which we choose analogously to the deterministic case), Castellana et al. (2019) showed that $f_\sigma \in [0.1, 0.5]$ results in the AMOC reaching an OFF-state within 100 years with probability at least $1/3$. In our case, as we are interested in millennial timescales, such noise strengths would lead the AMOC to collapse almost surely regardless of the applied meltwater fluxes in any experiment. For a better signal-to-noise ratio, we choose to use a weaker noise by setting $f_\sigma = 0.05$. Finally, to be able to later compare transition probabilities in the same $(\Delta t, P_{\mathrm{S}})$ and $(\Delta t_{\max}, P_{\mathrm{S}})$ parameter spaces as before, we must ensure that white noise takes effect at the same given time in every simulation. Hence, we will systematically start numerical integration at year $-3500$, corresponding to the minimal time at which the WAIS tipping is initiated. In the remainder of this section, transition probabilities are computed via Monte-Carlo sampling using 10000 trajectories.

First, we focus on transition probabilities when only the GIS meltwater flux forces the AMOC via equation (1). This is performed for two distinct values of $P_{\mathrm{N}} = 1000$ and 2000 years, where the former leads to an overshoot of the AMOC tipping point, while the latter does not. In Fig. 6, we show the time evolution of the cumulative transition probability $p_{\mathrm{GIS}}(t)$, standing for the probability that the AMOC transitions to an OFF state before time $t$. We observe that before the GIS collapse ($t < 0$), this probability increases almost linearly in time, solely due to the noise. From the moment that the GIS collapse is initiated ($t = 0$), the probability sharply increases as the value of $F_{\mathrm{N}}(t)$ brings the trajectory closer (or past) the border of the bistable region. This happens in both the cases where overshoot of the AMOC tipping point occurs or not although, in the case $P_{\mathrm{N}} = 2000$ years, the probability increases at a milder rate than for $P_{\mathrm{N}} = 1000$ years.

Second, using the same values of $P_{\mathrm{N}}$, we force the AMOC model with the full forcing given by equations (1)-(2) (or equations (1)-(3)). Integrating the system in time, this leads to a new cumulative probability of AMOC transition $p_{\mathrm{GIS,WAIS}}(t)$, interpreted in the same manner as $p_{\mathrm{GIS}}(t)$. To assess the magnitude of any stabilization effect originating from the WAIS meltwater flux, we use the following metric:

$$\delta p(t) \equiv \frac{p_{\mathrm{GIS,WAIS}}(t) - p_{\mathrm{GIS}}(t)}{p_{\mathrm{GIS}}(t)} * 100, \tag{6}$$

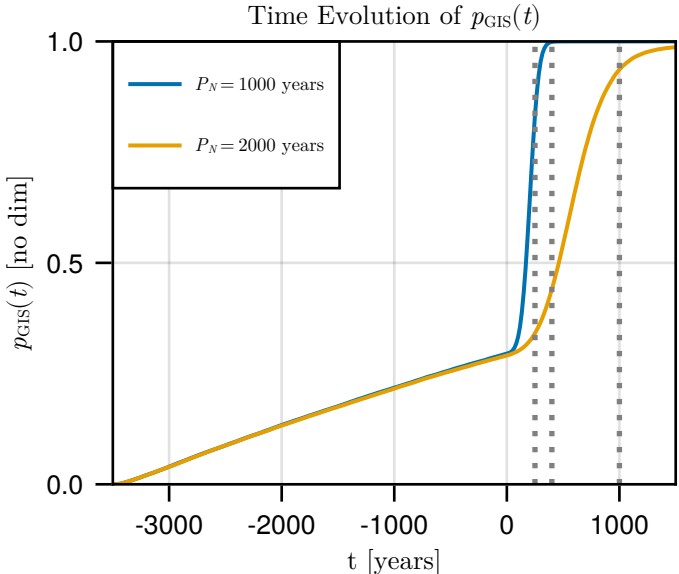

**Figure 6.** Time evolution of the cumulative probability of AMOC transition $p_{\mathrm{GIS}}(t)$ for $P_{\mathrm{N}} = 1000$ (blue) and $2000$ (yellow) years, using $f_{\sigma} = 0.05$ in both cases. Vertical dotted lines represent the three different times considered in Fig. 7. Probabilities were computed via Monte-Carlo sampling for $10000$ trajectories.

hereafter called relative probability change. It represents the percentage by which the cumulative transition probability varies with respect to the situation in which the GIS alone forces the AMOC. In other words, at any given time $t$, a negative value of
240    $\delta p(t)$ indicates a stabilization induced by the WAIS collapse.

In Fig. 7.a, $P_{\mathrm{N}} = 1000$ years was used, such that the AMOC tipping point is to be overshot everywhere except in the no-overshoot region (bounded by the red curve). The relative probability change is represented in both the $(\Delta t, P_{\mathrm{S}})$ parameter space (left column) and the $(\Delta t_{\max}, P_{\mathrm{S}})$ parameter spaces (right column) at three different times, namely at year $250$, $400$ and $1000$ (up to down, also drawn as dotted vertical lines in Fig. 6). At year $250$, $\delta p$ reaches -63%. While the overshoot of the
245    AMOC tipping point has not occurred yet (and does not before year $280$), it is interesting to note that the regions of strongest probability decrease are consistent in shape with the no-overshoot region in both parameter spaces, although with a clear shift, which will be discussed at the end of this section. At year $400$, only a small region of lower probability remains and gradually disappears as time increases, as can be seen at year $1000$, where $\delta p$ is uniformly close to zero. Results of the experiment using $P_{\mathrm{N}} = 2000$ years, meaning that there is no region in the $(\Delta t, P_{\mathrm{S}})$ parameter space where the AMOC tipping point is
250    ever overshot, are presented in Fig. 7.b. In this case, $\delta p$ at year $250$ is already substantially negative in a wide region of the parameter space, and even more so at year $400$. Importantly, we see that the stabilization is still present and substantial on longer timescales, as seen at year $1000$. It is also interesting to note that the region of strongest probability decrease gradually becomes similar in shape to the previously found no-overshoot region as, here, no overshoot is ever to occur. However, as $t$ increases, the center of this region clearly shifts toward higher values of $\Delta t_{\max}$. This also happens in the case $P_{\mathrm{N}} = 1000$ years

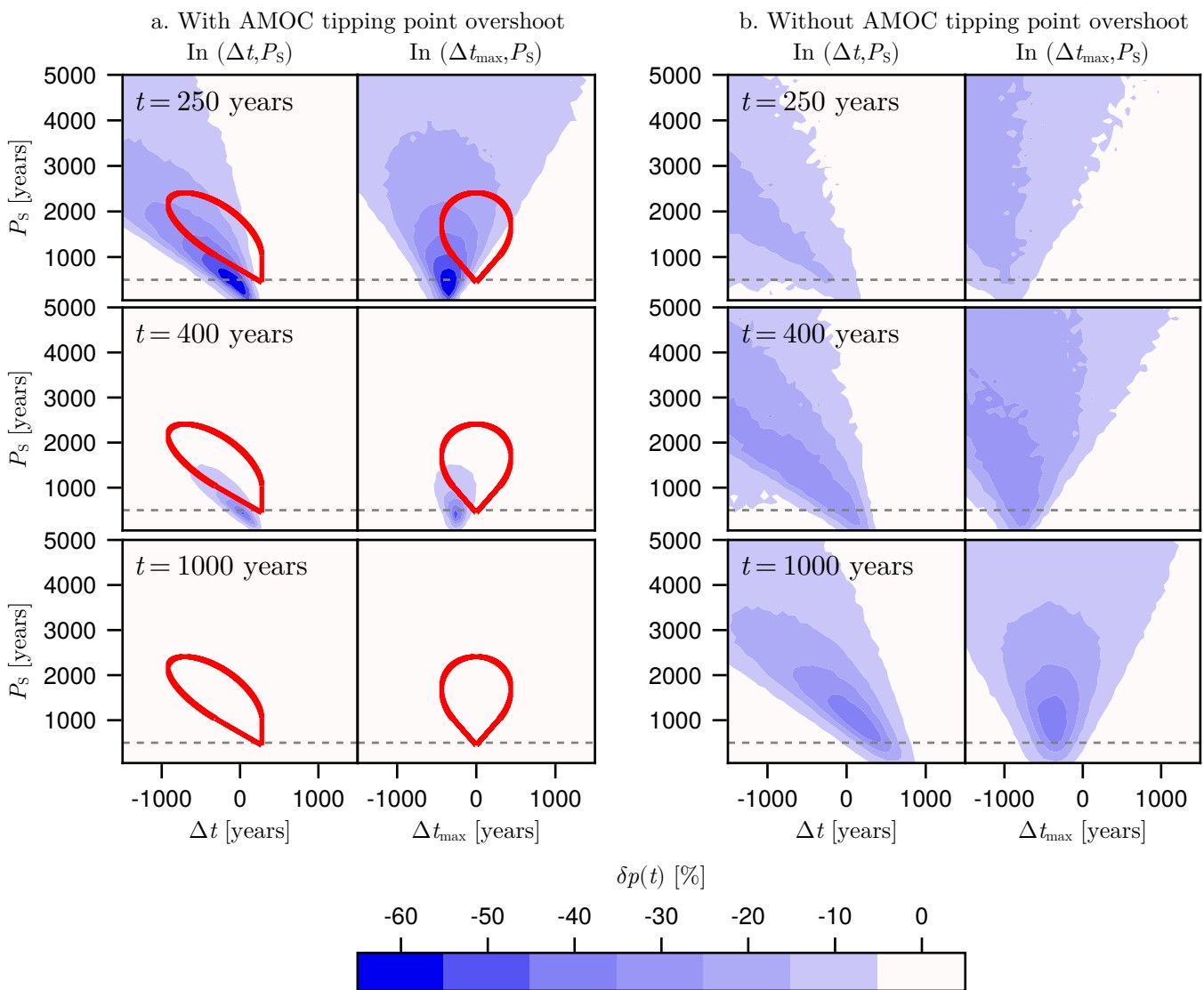

**Figure 7.** Evolution of the relative probability change $\delta p(t)$ (defined by equation (6)) in stochastic experiments using $f_\sigma = 0.05$. We use (a) $P_N = 1000$ years, in which case the tipping point of the AMOC is overshot outside of the no-overshoot region (bounded by the red curve) and (b) $P_N = 2000$ years, in which case the tipping point of the AMOC is never overshot. For both (a) and (b), results are drawn in the $(\Delta t, P_S)$ parameter space (left column) and in the $(\Delta t_{max}, P_S)$ parameter space (right column) at three different times, namely $t = 250, 400$ and 1000 years (top to bottom). The grey dashed line represents the minimal value of the WAIS tipping duration at $P_S = 500$ years proposed by Armstrong McKay et al. (2022). Each probability has been computed via Monte-Carlo sampling, using 10000 trajectories.

(Fig. 7.a), although less clearly. In summary, we find that the relative probability change can be substantially negative in both the case when an AMOC overshoot can occur or not. This probability decrease occurs for realistic values of the WAIS tipping duration $P_S$ (i.e. above the grey dashed line in Fig. 7.a-b). However, varying the forcing rate yields important quantitative differences in the magnitude of this stabilization effect, and on its specific timescale.

In Fig. 7.a, we found that the regions of strongest probability change and the no-overshoot region are similar in shape but not in location. Analogously to the results of section 4, it is tempting to also relate this shift to rate-induced effects. This can be verified by performing similar stochastic experiments for a lower value of the rate parameter $r$ (used as in section 4). In Fig. 8, we show the results obtained using $P_N = 1000$ years and $r = 0.25$. Again, to present results in the same $(\Delta t, P_S)$ and $(\Delta t_{max}, P_S)$ parameter space, we ensure that the noise is added at the same time in all experiments by starting integration at $t = -14000$ years. Hence, quantitative differences with respect to Fig. 7.a can be related to the fact that white noise acts during a much longer time, while the region in which $\delta p$ is most negative remains similar. Most importantly, we find that the no-overshoot region does not become a better approximation of the region of strongest probability decrease at any time, despite the low value used for the rate parameter $r$. This implies that the shift between the no-overshoot region and the region in which $\delta p$ is most negative cannot be explained (only) by rate-induced effects, but clearly involves noise-induced effects.

## 6    Summary and Discussion

In this study, the stability of the AMOC was explored in numerical simulations involving meltwater fluxes from both polar ice sheets. Although in a different experimental setup, the same qualitative results of Sinet et al. (2023) were verified, where the occurrence of an AMOC tipping depends on the timescale of both ice sheet collapses, and the delay between those collapses. In a worst case scenario where the GIS fully collapses in 1000 years (as suggested by Armstrong McKay et al. (2022)), deterministic results of section 3 indicated that a stabilizing influence of the WAIS collapse is strong enough to stabilize the AMOC. Notably, this stabilization can occur for a relatively fast WAIS collapse lasting up to 1600 years, corresponding to the lower projections of WAIS tipping durations (Armstrong McKay et al., 2022). Regarding the AMOC stabilization, the use of two different parametrizations of WAIS tipping trajectories informed us on the existence of some optimal scenario for the tipping of ice sheets, where the peak of WAIS meltwater flux occurs about 150 years before the peak of GIS meltwater flux.

In section 4, a detailed investigation of the AMOC bifurcation structure allowed to identify the no-overshoot region in the forcing parameter space, defining WAIS tipping scenarios for which the tipping point of the AMOC is never reached. This revealed that an AMOC collapse could be induced without leaving the bistable regime at any time, as well as the existence of safe-overshoot trajectories. Such rate-induced effects occur when ice sheets collapse on a timescale which is comparable to the recovery timescale of the AMOC, whose high value originates from the slow dynamics of the pycnocline depth. The existence of those rate-induced effects was confirmed by artificially slowing down the rate of variation of meltwater input, yielding results that became more consistent with bifurcation-induced critical transitions. This analysis allows us to find that the previously found optimal delay between the peaks of both meltwater fluxes is highly rate dependant.

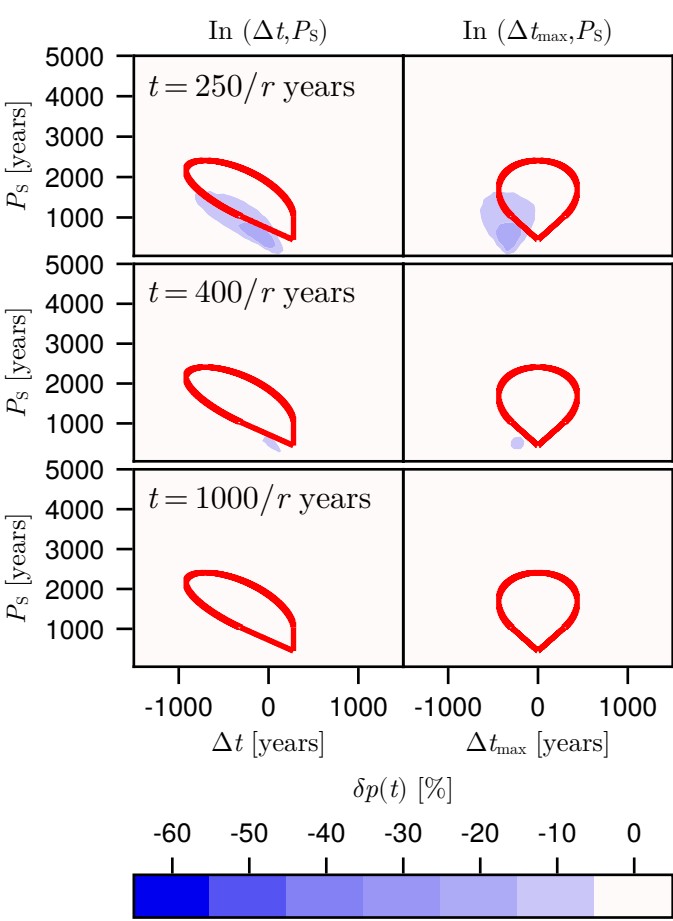

**Figure 8.** Same as Fig. 7.a, using a lower value of the rate parameter $r = 0.25$. Notably, the three values of $t$ used in Fig. 7.a are here divided by the rate parameter $r$. Each probability has been computed via Monte-Carlo sampling using 10000 trajectories.

Under stochastic noise in the surface freshwater flux (section 5), the stabilization effect of WAIS meltwater fluxes remains important. In contrast to the deterministic context (section 3), a drastic decrease of the AMOC transition probability is now possible in the case of slower WAIS collapse of up to 5000 years, spanning about the lower half of the range of WAIS tipping durations suggested by Armstrong McKay et al. (2022). On the one hand, we investigated the worst case scenario in which the GIS fully collapses in 1000 years, such that an overshoot of the AMOC tipping point can occur. In this case, we found that some WAIS trajectories result in a substantial decrease of the cumulative AMOC tipping probability (of up -63%), although remaining important only on the centennial timescale. On the other hand, we considered cases implying a comparatively slower GIS tipping lasting 2000 years, such that the tipping point of the AMOC is never overshot. In this case, we found that a similar probability decrease can persist way longer in time, on the millennial timescale. Finally, we were able to demonstrate that,

due to noise-induced effects, ice sheet tipping scenarios yielding the strongest stabilization cannot be accurately predicted by bifurcation analysis. Yet, similarities between the region of strongest stablization and the no-overshoot region motivates further investigation via the tools of stochastic dynamics (e.g, as in Berglund and Gentz, 2006).

Clearly, the scope of our results is limited by the conceptual nature of the model used, as was the case in Sinet et al. (2023). On the one hand, ice sheet melting trajectories were purposefully chosen to be minimalistic, and important processes were omitted. First, the cooling of the northern polar region induced by an AMOC weakening (Jackson et al., 2015; van Westen et al., 2024) was not included, and may render the AMOC tipping less likely via inhibition of a GIS collapse. Second, the warming of the Southern Hemisphere subsequent to an AMOC weakening (Jackson et al., 2015; Stouffer et al., 2006) was not considered, and could result in a WAIS collapse on a faster timescale. This could imply a shift of the time delay between the peak of both ice sheet meltwater flux towards negative values, thus either pushing this delay closer to or further away from its optimal value for a stabilization event to occur. On the other hand, only part of the AMOC dynamics is represented in a box model. In a more detailed setup, our quantitative results regarding rate and noise-induced effects would most likely be challenged and vary among different models. Indeed, comprehensive models display an important range of sensitivities to northern (Jackson et al., 2023) and southern (Swart et al., 2023) freshwater fluxes. Furthermore, the AMOC tipping point varies substantially in different Earth system Models of Intermediate Complexity (EMICs) (Rahmstorf et al., 2005), and was only recently found in a Global Climate Model (van Westen and Dijkstra, 2023). A direct extension of this study would be to investigate how the results compare to those obtained in EMICs which, as in our case, can be obtained using only prescribed meltwater fluxes.

Nonetheless, the existence of such important rate and noise-induced effects at the conceptual level informs us on the limits of an analysis focused solely on bifurcation-induced transitions. It suggests that maintaining meltwater fluxes under some critical value might not suffice for ensuring the AMOC stability. Yet, the significant stabilizing impact of a WAIS collapse on the AMOC stability as found in Sinet et al. (2023) is consistently found in a different conceptual AMOC model, and motivates further investigations. Finally, the decisive role of forcing rates encourages efforts to produce accurate predictions of ice sheet melting under future global warming.

*Code availability.* Julia codes used for producing the results and plots have been archived within the open access repository Zenodo (Sinet, 2023)

## Appendix A: AMOC Box Model and Numerical Methods

The conceptual AMOC model represents the Atlantic Ocean as five distinct boxes (see Fig. 1) of constant temperature but time-varying salinity, and includes a dynamical representation of the pycnocline depth. The time evolution of salinities $S_i$ for $i \in \{\mathrm{t, ts, n, s, d}\}$ and the pycnocline depth D are given by differential equations identical to those used earlier (Cimatoribus et al., 2014; Castellana et al., 2019; Jacques-Dumas et al., 2023), except for the addition of time-dependent meltwater fluxes

$F_{N,S}(t)$ defined by equations (1)-(2) (or equations (1)-(3)), as follows

$$\frac{d(V_t S_t)}{dt} = q_s \left(\theta\left(q_s\right) S_{ts} + \theta\left(-q_s\right) S_t\right) + q_u S_d - \theta\left(q_n\right) q_n S_t + r_s \left(S_{ts} - S_t\right) + r_n \left(S_n - S_t\right) + \left(2E_s + F_N(t) + F_S(t)\right) S_0 \quad \text{(A1)}$$

$$\frac{d(V_{ts} S_{ts})}{dt} = q_{Ek} S_s - q_e S_{ts} - q_s \left(\theta\left(q_s\right) S_{ts} + \theta\left(-q_s\right) S_t\right) + r_s \left(S_t - S_{ts}\right) \quad \text{(A2)}$$

$$\frac{d(V_n S_n)}{dt} = \theta\left(q_n\right) q_n \left(S_t - S_n\right) + r_n \left(S_t - S_n\right) - \left(E_s + E_a + F_N(t)\right) S_0 \quad \text{(A3)}$$

$$\frac{d(V_s S_s)}{dt} = q_s \left(\theta\left(q_s\right) S_d + \theta\left(-q_s\right) S_s\right) + q_e S_{ts} - q_{Ek} S_s - \left(E_s - E_a + F_S(t)\right) S_0 \quad \text{(A4)}$$

$$\left(A + \frac{L_{x_A} L_y}{2}\right) \frac{dD}{dt} = q_u + q_{Ek} - q_e - \theta\left(q_n\right) q_n \quad \text{(A5)}$$

$$S_0 V_0 = V_n S_n + V_d S_d + V_t S_t + V_{ts} S_{ts} + V_s S_s, \quad \text{(A6)}$$

where $\theta(x)$ stands for the Heaviside function. Those include wind driven transports $r_{s,n}$ originating from sub-tropical gyres, and the following volume transports

$$q_{Ek} = \frac{\tau L_{xS}}{\rho_0 \left|f_S\right|} \quad \text{(A7)}$$

$$q_e = A_{GM} \frac{L_{xA}}{L_y} D \quad \text{(A8)}$$

$$q_s = q_{Ek} - q_e \quad \text{(A9)}$$

$$q_n = \eta \frac{\rho_n - \rho_{ts}}{\rho_0} D^2 \quad \text{(A10)}$$

$$q_u = \frac{\kappa A}{D}, \quad \text{(A11)}$$

where densities $\rho_i$ for $i \in \{n, ts\}$ are given by

$$\rho_i = \rho_0 \left(1 - \alpha\left(T_i - T_0\right) + \beta\left(S_i - S_0\right)\right), \quad \text{(A12)}$$

and volumes are given by

$$V_t = AD \quad \text{(A13)}$$

$$V_{ts} = \frac{L_{xA} L_y}{D} \quad \text{(A14)}$$

$$V_d = V_0 - V_t - V_{ts} - V_n - V_s. \quad \text{(A15)}$$

The different constants used in the model are provided in Table A1. Thourghout the study, numerical time integration is performed using the Julia package DifferentialEquations.jl (Rackauckas and Nie, 2017). The Runge-Kutta algorithm of order 4 is used for integration of the deterministic model, while the SOSRI algortihm (best described in Rackauckas and Nie (2020)) is used for integration of the stochastic model. Finally, the Julia package BifurcationKit.jl (Veltz, 2020) is used for numerical continuation.

**Table A1.** Constants used for the AMOC model

| Parameters of the AMOC model | | |
|---|---|---|
| $V_0$ | $3 \times 10^{17}$ m$^3$ | Total volume of the basin |
| $V_n$ | $3 \times 10^{15}$ m$^3$ | Volume of the northern box |
| $V_S$ | $9 \times 10^{15}$ m$^3$ | Volume of the southern box |
| $A$ | $1 \times 10^{14}$ m$^2$ | Horizontal area of the Atlantic pycnocline |
| $L_{xA}$ | $1 \times 10^7$ m | Zonal extent of the Atlantic Ocean at its southern end |
| $L_y$ | $1 \times 10^6$ m | Meridional extent of the frontal region of the Southern Ocean |
| $L_{xS}$ | $3 \times 10^7$ m | Zonal extent of the Southern Ocean |
| $\tau$ | $0.1$Nm$^{-2}$ | Average zonal wind stress amplitude |
| $A_{GM}$ | $1700$ m$^2$ s$^{-1}$ | Eddy diffusivity |
| $f_S$ | $-10^{-4}$ s$^{-1}$ | Coriolis parameter |
| $\rho_0$ | $1027.5$ kg m$^{-3}$ | Reference density |
| $\kappa$ | $10^{-5}$ m$^2$ s$^{-1}$ | Vertical diffusivity |
| $S_0$ | $35$psu | Reference salinity |
| $T_0$ | $5$ K | Reference temperature |
| $T_n$ | $5$ K | Temperature of the northern box |
| $T_{ts}$ | $10$ K | Temperature of the box "ts" |
| $\eta$ | $3 \times 10^4$ m s$^{-1}$ | Hydraulic constant |
| $\alpha$ | $2 \times 10^{-4}$ K$^{-1}$ | Thermal expansion coefficient |
| $\beta$ | $8 \times 10^{-4}$psu$^{-1}$ | Haline contraction coefficient |
| $r_S$ | $1 \times 10^7$ m$^3$ s$^{-1}$ | Transport by the southern subtropical gyre |
| $r_n$ | $5 \times 10^6$ m$^3$ s$^{-1}$ | Transport by the northern subtropical gyre |
| $E_s$ | $0.17 \times 10^6$ m$^3$ s$^{-1}$ | Symmetric freshwater flux |

## Appendix B: No-Overshoot Zone

As justified in section 4, meltwater forcing trajectories will never bring the AMOC model outside of its bistable region if condition (4) is verified for all $t \in \mathbb{R}$. Given a set value of $P_N > 0$, such trajectories (if they exist) are found in the $(\Delta t, P_S)$ and $(\Delta t_{\max}, P_S)$ parameter spaces by analysis, yielding the no-overshoot region. In what follows, computations are performed in the $(\Delta t_{\max}, P_S)$ parameter space. We begin by noting that $F_N(t)$ overshoots the critical value $F_{N,c}$ if and only if

$$3V_N - 2P_N F_{N,c} > 0, \tag{B1}$$

without what condition (4) is trivially verified in all cases. In the case where (B1) holds, the overshoot occurs during the interval $]t_{o-}, t_{o+}[$, where

$$t_{o-} = -\frac{P_N^3}{6V_N}\sqrt{\frac{3V_N(3V_N - 2P_N F_{N,c})}{P_N^4}} + \frac{P_N}{2} \tag{B2}$$

and

$$t_{o+} = \frac{P_N^3}{6V_N}\sqrt{\frac{3V_N(3V_N - 2P_N F_{N,c})}{P_N^4}} + \frac{P_N}{2}. \tag{B3}$$

Now, by virtue of the positivity of $F_S(t)$, a necessary condition for condition (4) to be satisfied is that $F_S(t)$ does not vanish for any $t$ in $]t_{o-}, t_{o+}[$. In other words, $]t_{o-}, t_{o+}[$ must be fully contained in $\left[\Delta t_{\max} - \frac{P_S - P_N}{2}, \Delta t_{\max} + \frac{P_S + P_N}{2}\right]$, which is true if and only if

$$|\Delta t_{\max}| \leq t_{o+} - \frac{P_S + P_N}{2}. \tag{B4}$$

In this case, it remains only to ensure that the second degree polynomial

$$G(t) = F_N(t) - \delta F_S(t) - F_{N,c} \tag{B5}$$

$$= -\frac{6V_N}{P_N^3}t(t - P_N) + \frac{\delta}{4}\frac{6V_S}{P_S^3}\left(t - \Delta t_{\max} + \frac{P_S - P_N}{2}\right)\left(t - \Delta t_{\max} - \frac{P_S + P_N}{2}\right) - F_{N,c} \tag{B6}$$

does not overshoot zero within the time interval interval $]t_{0-}, t_{0+}[$. Again, by virtue of the positivity of $F_S(t)$, this is true if and only if one of the three following criteria holds

- $\frac{d^2 G(t)}{dt^2}$ is always positive or zero. This implies that $G(t)$ is majored by zero within $]t_{0-}, t_{0+}[$, and is true if and only if

$$P_S \leq P_N \sqrt[3]{\frac{\delta V_S}{4V_N}}; \tag{B7}$$

- $\frac{d^2 G(t)}{dt^2}$ is always strictly negative, but $G(t)$ does not overshoot zero for any value of $t$. It is true if and only if the discriminant of $G(t)$ is negative or zero, translating into

$$48P_S^4 V_N^2 + 3\delta^2 P_N^4 V_S^2 + 8\delta F_{N,c}P_N^4 P_S V_S + 48\delta P_N V_N P_S V_S (\Delta t_{\max})^2$$
$$- 32F_{N,c}P_N V_N P_S^4 - 12\delta P_N^3 V_N P_S V_S - 12\delta P_N V_N P_S^3 V_S \leq 0 \tag{B8}$$

- $\frac{d^2 G(t)}{dt^2}$ is always strictly negative and $G(t)$ overshoots zero, but only in a finite region outside of $]t_{0-}, t_{0+}[$. This is equivalent to requiring that the maximum of $G(t)$ is found outside of $]t_{0-}, t_{0+}[$, which is true if and only if

$$2t_{o+} - P_N \leq 2\delta P_N^3 V_S \left|\frac{\Delta t_{\max}}{4V_N P_S^3 - \delta V_S P_N^3}\right|. \tag{B9}$$

To summary, given a set value of $P_N > 0$ for which $F_N(t)$ overshoots $F_{N,c}$ (meaning that condition (B1) holds), forcing trajectories verifying condition (4) may exist. Those verify (B4), along with either criteria (B7),(B8) or (B9). We note that, as

all those condition are invariant under the transformation $\Delta t_{\mathrm{max}} \rightarrow -\Delta t_{\mathrm{max}}$, the no-overshoot region yields an axial symmetry around $\Delta t_{\mathrm{max}} = 0$. Finally, this region can be described in the $(\Delta t, P_{\mathrm{S}})$ parameter space, simply by substituting $\Delta t = \Delta t_{\mathrm{max}} - \frac{P_{\mathrm{S}} - P_{\mathrm{N}}}{2}$ in each condition.

*Author contributions.* SS carried out the analysis and drafted the paper. PA, ASvdH and HAD advised on the study. All authors contributed to the submitted paper.

*Competing interests.* No competing interests are present.

*Acknowledgements.* This project has received funding from the European Union's Horizon 2020 research and innovation programme under the Marie Skłodowska-Curie Grant Agreement No. 956170 (CriticalEarth). A.S. vdH acknowledges funding by the Dutch Research Council (NWO) through the NWO-Vici project 'Interacting climate tipping elements: When does tipping cause tipping?' (project VI.C.202.081).

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
