# Peer review of "AMOC Stability Amid Tipping Ice Sheets: The Crucial Role of Rate and Noise"

_EGUsphere, 2023_

## Author Response (AR1)

**Revision of the manuscript "AMOC Stability Amid Tipping Ice Sheets: The Crucial Role of Rate and Noise"**

**Sacha Sinet, Peter Ashwin, A.S. Von der Heydt and H.A. Dijkstra**

On behalf of all the authors, we express once more our sincere gratitude towards both reviewers, as their comments allowed for substantially improving the quality and clarity of our manuscript. In this revised version, each comment has been carefully considered and answered to.

This document is an exhaustive list of all changes applied, where all line references (L#) point to the trackchanges version of the manuscript. In a first time, we list some general changes which do not apply only to one specific comment:

- Fig. 1.b and its caption have been removed for redundancy, as the same information is now part of Fig. 2.a-b. Also, to improve the contextualisation of our results, we added an horizontal dashed line at the minimal value of the WAIS tipping duration at $P_S = 500$ years (Armstrong McKay et al., 2022), on Fig. 2.c-d and 7.a-b;

- for clarity and consistency, we systematically replaced the term "no-tipping region" by "stabilization region". This also allows to emphasise on the stabilization effect, which is a key result of the paper;

- some minor typos or formulation issues have been corrected. Notably, we changed notation for boxes of the model to avoid confusion with the time symbol $t$ (e.g., box $t$ become box t).

- the citation of Wunderling et al. (2024), which has been published in the meantime, has been updated. Citation to Rosier et al. (2021) and Westen, Kliphuis, and Henk A. Dijkstra (2024) have been added.

In a second time, on top of the previously shared answer to the reviewers, we provide a more precise "Changes in text" section (in blue), which have been the result of RC1 in "Answer to RC1" and those which resulted RC2 in "Answer to RC2".

Sincerely yours,

the authors

**Answer to RC1**

We thank the reviewer for this overall positive feedback and insightful comments. Below, we provide individual answers to each comment, along with planned modifications of the manuscript.
* * *
*This study investigates the role of (1) GIS and WAIS meltwater forcing rate, (2) stochastic noise on meltwater forcing, on AMOC collapse using a 5-box conceptual AMOC model. The paper is technically sound and confirms the idea that the WAIS has a stabilizing effect on the AMOC (proposed by previous studies) even under the influence of forcing rate and noise.*

*The paper is worthy of publication in ESD, however additional analysis/revisions are needed. Please see the comments below.*

*Additional analysis with realistic meltwater flux forcing: I propose an additional analysis/discussion on the AMOC collapse at the realistic range of forcing parameters (Eq (1) and Eq (2)). First, please provide a realistic range of the forcing parameters in the context of a palaeoclimate event (e.g., MWP-1A) or future climate change (e.g., CMIP6 SSP5-8.5 scenario). Then, discuss the AMOC collapse behavior at this realistic range of meltwater flux. If this specific past/future event is not applicable to the forcing scenario used in this paper (which assumes full melting of GIS and WAIS), I suggest performing an additional experiment. This would provide practical insights into the AMOC collapse in the past/future. Also, it would shore up a weakness of the paper (the weakness of using a highly idealized model with conceptual forcing).*

**Reply:** our manuscript focuses on the qualitative aspects of rate and noise-induced effects on the AMOC stability. However, it is motivated by future climate change and, as such, we ensured that the different forcing parameters used were consistent with Armstrong McKay et al. (2022), the most comprehensive and up-to-date reference regarding future tipping events. In this paper, a most likely scenario is also provided, wherein the GIS collapses in 10,000 years and the WAIS in 2,000 years. In our deterministic model, such a GIS forcing is far from being strong enough to result in an AMOC tipping. Therefore, in this most likely scenario, the AMOC remains stable independently of the applied WAIS meltwater flux.

In our paper, a collapse duration of 1,000 years for the GIS was used in the deterministic case, which is the lower limit of the range of 1,000 to 13,000 years proposed by Armstrong McKay et al. (2022). Hence, as suggested around line 110, this must be interpreted as a worst case scenario. This is further motivated by Aschwanden et al. (2019), where a high emission scenario (extended RCP8.5) yields a collapse of the GIS at the millennial timescale. In such a scenario, all the range of plausible WAIS tipping points are exceeded in less than a century, resulting in a negligible time delay between the two ice sheet tipping events (i.e. $\Delta t \approx 0$). Therefore, in this worst case scenario, a collapse duration of WAIS of approximately 500 to 1,200 years results in a stabilization of the AMOC (Fig. 2a). Such considerations can also be made in the stochastic experiments, although in terms of the probability of AMOC collapse.

While it is true that those results could also provide insight into paleoclimatic events (e.g. MWP-1A), we did not discuss such aspects as the conceptual AMOC model of Cimatoribus,

Drijfhout, and Henk A. Dijkstra (2014) is built and parametrised to represent a present-day AMOC. As such, we prefer to avoid speculating in this direction.

**Changes in text:** contextualisation of our results in terms of future climate change has been added in section 3, L153-161:

"Those results can be interpreted in the context of future climate change. In Armstrong McKay et al. (2022), plausible values for the GIS tipping duration $P_N$ are given in the range $[1000, 15000]$ years, with a most likely value of 10000 years. In our model, this most likely scenario results in the AMOC to remain stable regardless of the applied WAIS meltwater flux. Instead, $P_N = 1000$ years can be thought of as a worst case scenario. This is further motivated by the modelling study of Aschwanden et al. (2019), in which a GIS collapsing on the millenial timescale was found under a RCP8.5 scenario. Under such conditions, the whole range of plausible WAIS tipping points of $[1.5, 3.0]°C$ of global warming above pre-industrial levels (Armstrong McKay et al., 2022) would be overshot in less than a century, resulting in a negligible value of the time delay between ice sheet tipping events ($\Delta t \approx 0$ years). Hence, in such a worst case scenario, a stabilization would occur for values of $P_S$ between approximately 500 and 1300 years (see the representative trajectory in Fig. 2.b)."

Also, we added a horizontal dashed line at the minimal realistic values of the WAIS tipping duration (at $P_S = 500$ years, from Armstrong McKay et al. (2022)) in Fig. 2.c-d and 7.a-b, referred to in lines 136-137 (and analogously in L270-271):

"... In particular, we find that the stabilization region exists for realistic values of the WAIS tipping duration $P_S$ (i.e. above the grey dashed line in Fig. 2.c-d)."
* * *
*Line 80: The authors consider the time delay between FN (representative of GIS collapse) and FS (representative of WIS collapse) forcing as a key parameter of the AMOC experiment. What is the physical motivation for setting a time delay between them? What is a realistic range of time delay in the context of palaeoclimate and future climate change?*

**Reply:** given the uncertainty on both ice sheet tipping points and the uncertainty in future climate warming trajectories, a wide range of delays between ice sheet tipping events is in principle possible.

For example, considering a global warming scenario in which global temperature increases from 1.1°C to 3.0°C above preindustrial level in 1,500 years, the WAIS would begin to collapse 1,500 years after the GIS if their respective tipping points are at 3.0°C and 1.1°C. Conversely, if those tipping points are inverted, the WAIS would begin to collapse 1,500 years before the GIS.

**Changes in text:** the motivation for considering varying time delays and tipping timescales is now part of a new paragraph in section 2, L110-116:

"Hence, the forcing (1)-(2) (or equivalently (1)-(3)) allows for conceptually capturing a full collapse of both ice sheets while spanning a wide range of possible tipping durations and delays, which is motivated by the uncertainty in ice sheet tipping points, tipping timescales and global warming trajectories (Armstrong McKay et al., 2022), as well as the important role of those parameters found in Sinet, Heydt, and H. A. Dijkstra (2023). In the remainder of this document, we consider the interval $[50, 5000]$ years for the WAIS tipping duration $P_{\mathrm{S}}$, encompassing about the lower half of the interval $[500, 13000]$ years proposed by Armstrong McKay et al. (2022). Also, to explore the impact of different ice sheet collapse trajectories, we use the delay interval $[-1500, 1500]$ years for both $\Delta t$ and $\Delta t_{\max}$."
* * *
*Section 3: Please show the figure that shows together the case of WAIS included and the non-included case for the AMOC collapse (time series would be good). The WAIS-induced stabilization effect is an important key message of this paper, so the direct comparison of these two cases will improve the presentation of the paper (the current version of the figure set is not friendly to readers who are not familiar with the low-order AMOC modelling and dynamical systems theory).*

**Reply:** we thank the reviewer for this valuable suggestion.

**Changes in text:** time series of the ice sheet melwater fluxes and the overturning strength have been implemented in the manuscript as part of Fig. 2.a (WAIS non included and AMOC collapse) and 2.b (WAIS included and AMOC stabilization), adding the caption:

"(a-b) Representative trajectories of meltwater fluxes $F_{\mathrm{N,S}}(t)$ as given by the set of equations (1)-(2) (or equations (1)-(3)) and overturning strength $q_{\mathrm{n}}$. In (a), only a GIS collapse lasting $P_{\mathrm{N}} = 1000$ years forces the AMOC model, resulting in an AMOC tipping ($q_{\mathrm{n}} = 0$). In (b), both a GIS and a WAIS collapse force the AMOC model. Those last $P_{\mathrm{N}} = 1000$ years and $P_{\mathrm{S}} = 900$ years, respectively, and are initiated at the same time ($\Delta t = 0$ years, or equivalently $\Delta t_{\max} = -50$ years), resulting in an AMOC stabilization."

Those are referred to in L128 and 161. Note that, for redundancy, this resulted in the removal of Fig 1.b. Finally, the term "no-tipping region" has been systematically replaced by "stabilization region", emphasising on this key message of the paper.
* * *
*Limitation of the conceptual model: The 5-box AMOC model used in this study does not consider the AMOC impact on the WAIS melt. The model considers only a one-way influence from the WAIS melt to the AMOC. However, as the authors explained in the introduction, the collapse of AMOC would increase the Southern Hemisphere temperature and accelerate the WAIS melt, while decreasing the Northern Hemisphere temperature and decelerating the GIS melt. The discussion of this missing physics (which may be very important) should be explained in the paper (probably in Section 6).*

**Reply:** our experiments are based on meltwater forcing trajectories which have been simplified into parabolas, solely defined by their duration in time and initiation time, independently of the AMOC dynamics. While this does not allow for implementing such feedbacks, some of their impact can be discussed in light of our results as follows.

On one hand, cooling of the North Pole tends to inhibit a GIS tipping event, which renders an AMOC collapse less likely. On the other hand, warming of the southern ocean implies an earlier and/or faster WAIS collapse. Especially, it would result in an earlier maximum of the WAIS meltwater flux, implying a shift of the $\Delta t_{\max}$ forcing parameter towards negative values. In the deterministic experiment, we found that $\Delta t_{\max} \approx -150$ years is optimal to result in an AMOC stabilization. Hence, the consequences of this feedback are more nuanced, as it may facilitate or inhibit AMOC tipping if it drives $\Delta t_{\max}$ away from or towards this optimal value, respectively.

**Changes in text:** discussion of those feedbacks has been added in section 6, L318-327:

"Clearly, the scope of our results is limited by the conceptual nature of the model used, as was the case in Sinet, Heydt, and H. A. Dijkstra (2023). On the one hand, only part of the AMOC dynamics is represented in a box model. Hence, a direct extension of this study would be to investigate how the results compare to those obtained in a three-dimensional comprehensive global ocean model which, as in our case, can be obtained using only prescribed meltwater fluxes. On the other hand, ice sheet melting trajectories were purposefully chosen to be minimalistic, and important processes were omitted. First, the cooling of the northern polar region induced by an AMOC weakening (Jackson et al., 2015; Westen, Kliphuis, and Henk A. Dijkstra, 2024) was not included, and may render the AMOC tipping less likely via inhibition of a GIS collapse. Second, the warming of the Southern Hemisphere subsequent to an AMOC weakening (Jackson et al., 2015; Stouffer et al., 2006) was not considered, and could result in a WAIS collapse on a faster timescale. This could imply a shift of the time delay between the peak of both ice sheet meltwater flux towards negative values, thus either pushing this delay closer to or further away from its optimal value for a stabilization event to occur."
* * *
*L203: Which numerical scheme for stochastic differential equations is used to solve the weak noise case?*

**Reply:** the algorithm SOSRI was used. It is a stability-optimized adaptive integration algorithm for sotchastic differential equations, best described in Chris Rackauckas and Nie (2020). It is implemented in the Julia package DifferentialEquations.jl (Christopher Rackauckas and Nie, 2017), which was used throughout the study.

**Changes in text:** description of numerical methods and tools are now explicit in the appendix A, which was renamed "Appendix A: AMOC Box Model and Numerical Methods", L361-365:

"Thourghout the study, numerical time integration is performed using the Julia package DifferentialEquations.jl (Christopher Rackauckas and Nie, 2017). The Runge-Kutta algorithm of order 4 is used for integration of the deterministic model, while the SOSRI algortihm (best described in (Chris Rackauckas and Nie, 2020)) is used for integration of the stochastic model. Finally, the Julia package BifurcationKit.jl (Veltz, 2020) is used for numerical continuation."
* * *
*Abstract: The key results of the paper are summarized too much in the abstract, and do not give an immediate answer to the question likely to arise from the title (so, what is the role of the forcing rate and noise?). Please revise it.*

**Reply:** the answer provided by this manuscript is that both rate and noise-induced effects have substantial impact on which ice sheet collapse trajectories may or not result in an AMOC tipping.

**Changes in text:** to be more explicit about the key results of our study, we added the following sentences to the abstract, L10-14:

"... While a substantial stabilization is found in both cases, we find that rate and noise-induced effects have substantial impact on the AMOC stability, as those imply that leaving the AMOC bistable regime is neither necessary nor sufficient for the AMOC to tip. Also, we find that rate-induced effects tend to allow for a stabilization of the AMOC in cases where the peak of the West Antarctica ice sheet meltwater flux occurs before the peak of the Greenland ice sheet meltwater flux."

**Answer to RC2**

We thank the reviewer for those useful comments, and especially for motivating a more precise description of some aspects related to the AMOC model. Below, we address each comment individually and outline planned modifications to the manuscript.
* * *
*This study discusses the AMOC stability under meltwater forcing from both Greenland and West Antarctica. The paper is interesting and sound if we accept the conceptual representation of AMOC. However, I struggled when trying to put these discussions in the context of real ocean. It may help the readers by including more comparison with the real world in the revised manuscript.*

*In particular, I wonder whether the AMOC sensitivity to WAIS meltwater has been exaggerated too much in this toy model. In a recent paper (https://doi.org/10.1175/JCLI-D-22-0433.1), which uses a more realistic model to examine the overturning responses to meltwater fluxes, they found that the AMOC is rather insensitive to Antarctic meltwater at least on timescales of 150 years. I think the "overestimated" sensitivity of the AMOC to WAIS meltwater in this toy model is because they parameterize the AMOC strength using density differences between the North Atlantic (n) and Southern Ocean (ts), rather than the density differences between the North Atlantic and the mid-latitude box (t) – the latter appears more plausible by physics (e.g., https://journals.ametsoc.org/view/journals/phoc/42/10/jpo-d-11-0189.1.xml). Processes in the Southern Ocean could affect the subsurface stratification in the low-mid latitudes, but this connection likely occurs on millennial timescales.*

**Reply:** as suggested by the reviewer, it is to be expected that different models present different sensitivities to forcing, especially when comparing conceptual and comprehensive models. However, we note that the study of Li et al. (2023) and ours are not incompatible for at least the two following reasons:

1. The forcing used in Li et al. (2023) is very different. Namely, it is uniform in time, while ours assumes different parabolic profiles for both ice sheet meltwater fluxes;

2. Li et al. (2023) found a limited sensitivity to Antarctic freshwater fluxes on a present-day AMOC, which is also the case for the model used in this manuscript. For example, they found an increase of the AMOC strength of approximately 0.63 Sv (or 2.8%) for a constant Antarctic freshwater flux of 0.06 Sv (Fig. 9 in Li et al. (2023)). In our model, the steady state of the circulation in terms of the AMOC strength is increased by approximately 1 Sv (or 6.9%) when a WAIS meltwater flux of 0.08 Sv is applied (Fig 3.a).

On the parametrization of the AMOC strength (here taken as the downwelling strength $q_N$), we note that the box $ts$ does not stand for the southern ocean (which is rather represented by the box s), but for the southern thermocline, or the part of the Atlantic Ocean between the Antarctic Circumpolar Current and the southern tip of Africa. This choice of parametrising the downwelling strength $q_N$ using the density of the southern thermocline box $ts$ rather than than the thermocline box $t$ is further motivated in Section 2.1 of Cimatoribus, Drijfhout, and Henk A. Dijkstra (2014).

**Changes in text:** Considerations on the sensitivity of the AMOC model to WAIS meltwater fluxes has been added as supplementary material in section S1, comparing our model to the one used in Li et al. (2023). It is referred to in section 4, L170-172:

"Via a similar approach, we also find that the sensitivity of the overturning strength $q_n$ to WAIS meltwater fluxes in the reference ON state is higher but comparable to what was found in Li et al. (2023) using the coupled climate model GISS-E2.1-G (see section S1)."

For changes related to clarification on the use of box $ts$, see next comment.
* * *
*I also have a few questions regarding the configuration of the AMOC model.*

*1. Why including the box "ts"? If I read it correctly, the overturning between ts and t is the same as the overturning between s and ts or d and s.*

**Reply:** the addition of the southern thermocline box $ts$ is extensively motivated in Cimatoribus, Drijfhout, and Henk A. Dijkstra (2014). To summarize, it allows for computing the density gradient within the Atlantic basin, and represents a region in which the isopycnal slopes are greater than in other parts of the thermocline.

**Changes in text:** the role of the and signification of box ts is now explicit in section 2, L78-80 :

" This box model includes two thermocline boxes t and ts, where the latter represents a region in which the isopycnal slopes are greater than in other parts of the thermocline and allows for computing the density gradient within the Atlantic basin."
* * *
*2. It should be clarified that the meltwater flux is applied as virtual salt flux not as freshwater flux, which won't influence the salinity content in box "t".*

**Reply:** we thank the reviewer for this sensible remark. It is however clear that, whether it originates from freshwater or virtual salinity flux, any change of salinity in any of the boxes will ultimately impact the salinity in other locations via the dynamics of the model.

**Changes in text:** the use of virtual salinity fluxes is now explicit in L81 (and analogously in L88-89):

"...(represented through virtual salt fluxes)..."
* * *
*3. How is temperature evolved in each box? As the authors mentioned in their introduction, a perturbation to the overturning circulation also modifies the ocean's thermal structure, which necessarily will feedback to the system and may affect the AMOC stability.*

**Reply:** temperatures are fixed in this model. Note that the temperature of boxes $n$, $ts$ and the reference temperature $T_0$ (which are the only ones explicitly used in the model) are given in table A1.

**Changes in text:** the treatment of temperatures as fixed parameters is now explicit in Appendix A, L337-338:

"The conceptual AMOC model represents the Atlantic Ocean as five distinct boxes (see Fig. 1) of constant temperature but time-varying salinity, and includes a dynamical representation of the pycnocline depth"

**Bibliography**

Armstrong McKay, David I. et al. (Sept. 2022). "Exceeding 1.5°C global warming could trigger multiple climate tipping points". In: *Science* 377(6611). Publisher: American Association for the Advancement of Science, eabn7950. DOI: 10.1126/science.abn7950. URL: https://www.science.org/doi/10.1126/science.abn7950 (visited on 10/07/2022).

Aschwanden, Andy et al. (June 2019). "Contribution of the Greenland Ice Sheet to sea level over the next millennium". In: *Science Advances* 5(6). Publisher: American Association for the Advancement of Science, eaav9396. DOI: 10.1126/sciadv.aav9396. URL: https://www.science.org/doi/10.1126/sciadv.aav9396 (visited on 03/22/2024).

Cimatoribus, Andrea A., Sybren S. Drijfhout, and Henk A. Dijkstra (Jan. 2014). "Meridional overturning circulation: stability and ocean feedbacks in a box model". en. In: *Climate Dynamics* 42(1), pp. 311–328. ISSN: 1432-0894. DOI: 10.1007/s00382-012-1576-9. URL: https://doi.org/10.1007/s00382-012-1576-9 (visited on 10/07/2022).

Jackson, L. C. et al. (Dec. 2015). "Global and European climate impacts of a slowdown of the AMOC in a high resolution GCM". en. In: *Climate Dynamics* 45(11-12), pp. 3299–3316. ISSN: 0930-7575, 1432-0894. DOI: 10.1007/s00382-015-2540-2. URL: http://link.springer.com/10.1007/s00382-015-2540-2 (visited on 10/13/2022).

Li, Qian et al. (May 2023). "Global Climate Impacts of Greenland and Antarctic Meltwater: A Comparative Study". EN. In: *Journal of Climate* 36(11). Publisher: American Meteorological Society Section: Journal of Climate, pp. 3571–3590. ISSN: 0894-8755, 1520-0442. DOI: 10.1175/JCLI-D-22-0433.1. URL: https://journals.ametsoc.org/view/journals/clim/36/11/JCLI-D-22-0433.1.xml (visited on 04/23/2024).

Rackauckas, Chris and Qing Nie (Sept. 2020). "Stability-Optimized High Order Methods and Stiffness Detection for Pathwise Stiff Stochastic Differential Equations". In: *2020 IEEE High Performance Extreme Computing Conference (HPEC)*. ISSN: 2643-1971, pp. 1–8. DOI: 10.1109/HPEC43674.2020.9286178. URL: https://ieeexplore.ieee.org/document/9286178 (visited on 03/25/2024).

Rackauckas, Christopher and Qing Nie (2017). "Differentialequations.jl–a performant and feature-rich ecosystem for solving differential equations in julia". In: *Journal of Open Research Software* 5(1), p. 15.

Rosier, Sebastian H. R. et al. (Mar. 2021). "The tipping points and early warning indicators for Pine Island Glacier, West Antarctica". English. In: *The Cryosphere* 15(3). Publisher: Copernicus GmbH, pp. 1501–1516. ISSN: 1994-0416. DOI: 10.5194/tc-15-1501-

2021. URL: `https://tc.copernicus.org/articles/15/1501/2021/` (visited on 01/24/2023).

Sinet, S., A. S. von der Heydt, and H. A. Dijkstra (2023). "AMOC Stabilization Under the Interaction With Tipping Polar Ice Sheets". en. In: *Geophysical Research Letters* 50(2). _eprint: https://onlinelibrary.wiley.com/doi/pdf/10.1029/2022GL100305, e2022GL100305. ISSN: 1944-8007. DOI: `10.1029/2022GL100305`. URL: `https://onlinelibrary.wiley.com/doi/abs/10.1029/2022GL100305` (visited on 01/20/2023).

Stouffer, R. J. et al. (Apr. 2006). "Investigating the Causes of the Response of the Thermohaline Circulation to Past and Future Climate Changes". EN. In: *Journal of Climate* 19(8). Publisher: American Meteorological Society Section: Journal of Climate, pp. 1365–1387. ISSN: 0894-8755, 1520-0442. DOI: `10.1175/JCLI3689.1`. URL: `https://journals.ametsoc.org/view/journals/clim/19/8/jcli3689.1.xml` (visited on 03/14/2023).

Veltz, Romain (July 2020). *BifurcationKit.jl*. URL: `https://hal.archives-ouvertes.fr/hal-02902346`.

Westen, René M. van, Michael Kliphuis, and Henk A. Dijkstra (Feb. 2024). "Physics-based early warning signal shows that AMOC is on tipping course". In: *Science Advances* 10(6). Publisher: American Association for the Advancement of Science, eadk1189. DOI: `10.1126/sciadv.adk1189`. URL: `https://www.science.org/doi/full/10.1126/sciadv.adk1189` (visited on 03/14/2024).

Wunderling, Nico et al. (Jan. 2024). "Climate tipping point interactions and cascades: a review". English. In: *Earth System Dynamics* 15(1). Publisher: Copernicus GmbH, pp. 41–74. ISSN: 2190-4979. DOI: `10.5194/esd-15-41-2024`. URL: `https://esd.copernicus.org/articles/15/41/2024/` (visited on 01/26/2024).

---

## Author Response (AR2)

**Minor revision of the manuscript "AMOC Stability Amid Tipping Ice Sheets: The Crucial Role of Rate and Noise"**

**Sacha Sinet, Peter Ashwin, A.S. Von der Heydt and H.A. Dijkstra**

We thank the reviewer for the positive evaluation of the revised manuscript.

As proposed by the editor, we have rewritten the discussion to clearly state that our quantitative results are likely to change in comprehensive models. This can be found in L309-316 of the trackchanges version of the manuscript:

'... On the other hand, only part of the AMOC dynamics is represented in a box model. In a more realistic setup, our quantitative results regarding rate and noise-induced effects would most likely be challenged and vary among different models. Indeed, comprehensive models display an important range of sensitivites to northern (Jackson et al., 2023) and southern (Swart et al., 2023) freshwater fluxes. Furthermore, the AMOC tipping point varies substantially in different Earth system Models of Intermediate Complexity (EMICs) (Rahmstorf et al., 2005), and was only recently found in a Global Circulation Model (Westen and Dijkstra, 2023). As a first step, a direct extension of this study would be to investigate how the results compare to those obtained in EMICs which, as in our case, can be obtained within a realistic computational time and using only prescribed meltwater fluxes.'

Note that references to Jackson et al. (2023); Swart et al. (2023); Rahmstorf et al. (2005) and Westen and Dijkstra (2023) were added to this version of the manuscript. Finally, a few typos have been corrected.

Sincerly,

The authors

**Bibliography**

Jackson, Laura C. et al. (Apr. 2023). "Understanding AMOC stability: the North Atlantic Hosing Model Intercomparison Project". English. In: *Geoscientific Model Development* 16(7). Publisher: Copernicus GmbH, pp. 1975–1995. ISSN: 1991-959X. DOI: `10.5194/gmd-16-1975-2023`. URL: `https://gmd.copernicus.org/articles/16/1975/2023/` (visited on 05/22/2024).

Rahmstorf, Stefan et al. (2005). "Thermohaline circulation hysteresis: A model intercomparison". en. In: *Geophysical Research Letters* 32(23). _eprint: https://onlinelibrary.wiley.com/doi/pdf/10.1029/ ISSN: 1944-8007. DOI: `10.1029/2005GL023655`. URL: `https://onlinelibrary.wiley.com/doi/abs/10.1029/2005GL023655` (visited on 10/07/2022).

Swart, Neil C. et al. (Dec. 2023). "The Southern Ocean Freshwater Input from Antarctica (SOFIA) Initiative: scientific objectives and experimental design". English. In: *Geoscientific Model Development* 16(24). Publisher: Copernicus GmbH, pp. 7289–7309. ISSN: 1991-959X. DOI: `10.5194/gmd-16-7289-2023`. URL: `https://gmd.copernicus.org/articles/16/7289/2023/` (visited on 05/22/2024).

Westen, René M. van and Henk A. Dijkstra (2023). "Asymmetry of AMOC Hysteresis in a State-Of-The-Art Global Climate Model". en. In: *Geophysical Research Letters* 50(22). _eprint: https://onlinelibrary.wiley.com/doi/pdf/10.1029/2023GL106088, e2023GL106088. ISSN: 1944-8007. DOI: `10.1029/2023GL106088`. URL: `https://onlinelibrary.wiley.com/doi/abs/10.1029/2023GL106088` (visited on 04/10/2024).